# Photodynamics of the Molecular Ruby [Cr(ddpd)_2_]^3+^

**DOI:** 10.3390/molecules28041668

**Published:** 2023-02-09

**Authors:** J. Patrick Zobel, Hanna Radatz, Leticia González

**Affiliations:** Institute of Theoretical Chemistry, Faculty of Chemistry, University of Vienna, Währinger Straße 17, 1090 Vienna, Austria

**Keywords:** excited-state dynamics, transition-metal complexes, near-infrared emitter, molecular ruby, chromium, surface hopping, metal-centered states, luminescence

## Abstract

The introduction of strong-field ligands can enable luminescence in first-row transition-metal complexes. In this way, earth-abundant near-infrared emitters can be obtained using early 3d metals. A prime example is the molecular ruby [Cr(ddpd)_2_]^3+^ (ddpd = N,N′-dimethyl-N,N′-dipyridin-2-ylpyridine-2,6-diamine) that can achieve high phosphorescence quantum yields at room temperature in aqueous solution. To understand these remarkable properties, here, we simulate its photodynamics in water using trajectory surface hopping on linear vibronic coupling potentials parametrized from multiconfigurational CASSCF/CASPT2 calculations. We find that after excitation to the second absorption band, a relaxation cascade through metal-centered states occurs. After an initial back-and-forth intersystem crossing with higher-lying doublet states, the complex relaxes through a manifold of quartet metal-centered states to the low-lying doublet metal-centered states which are responsible for the experimentally observed emission. These electronic processes are driven by an elongation of the Cr–ligand bond lengths as well as the twisting motion of the trans-coordinated pyridine units in the ddpd ligands. The low-lying doublet states are reached within 1–2 ps and are close in geometry to the doublet minima, thus explaining the high phosphorescence quantum yield of the molecular ruby [Cr(ddpd)_2_]^3+^.

## 1. Introduction

The rich photochemistry and photophysics of heavy transition-metal complexes [1] are usually enabled by their ability to access long-lived electronically excited states after photoexcitation. This long lifetime opens the possibility to harvest their excess energy, either directly through luminescence, e.g., in biomedical applications [2,3], or indirectly, e.g., when the hole and electrons in the excited-state electron density are extracted from the complexes in solar-energy conversion technologies [4]. Establishing the same applications within first-row transition-metal complexes is more challenging, motivating many current research efforts [5,6]. This challenge stems from the weaker ligand-field splitting that 3d transition-metal complexes experience compared to their heavier 4d and 5d analogues. The weaker splitting results in low-lying metal-centered electronic states which can provide efficient nonradiative relaxation pathways to the electronic ground state, thus reducing the lifetime of the useful electronic excited states. Overcoming this challenge though is highly rewarding as the earth-abundance of 3d transition metals makes applications more economic and sustainable.

Along these lines, a growing number of luminescent first-row transition-metal complexes has been reported [7,8]. They can be broadly classified into two categories depending on the nature of their emissive state. Charge-transfer emissive states are found mostly for $d^{5}$–$d^{10}$ metals including iron(III) [9], cobalt(III) [10], nickel(II) [11] and copper(I) [12], which emit in the visible spectral range. Longer emission wavelengths reaching the near-infrared regime can be obtained in chromium(III) and vanadium(III) complexes that possess metal-centered emissive states [13]. The prime example of the latter class of complexes is thereby given by [Cr(ddpd)_2_]^3+^ (ddpd = N,N′-dimethyl-N,N′-dipyridine-2-ylpyridine-2,6-diamine), see Figure 1a. The complex has been nicknamed the molecular ruby in analogy to emissive Cr(III) ions surrounded by six O${}^{2-}$ ions in solid ruby [14].

[Cr(ddpd)_2_]^3+^ displays near-infrared phosphorescence in aqueous solution at 778 and 740 nm with a quantum yield of 11% that occurs from the almost degenerate ^2^T_1_ and ^2^E electronic terms [15]. The high phosphorescence efficiency is achieved through the combination of the electronic and geometric properties of the ddpd ligand. Both the strong σ-donor and poor π-acceptor qualities of the coordinating pyridine units as well as the large bite angle close to 90${}^{\circ }$ of the tridentate ddpd scaffold lead to a large ligand-field splitting in the electronic states of [Cr(ddpd)_2_]^3+^. Phosphorescence in octahedral chromium(III) complexes is usually quenched through a nonradiative relaxation that includes (back) intersystem crossing from the ^2^T_1_ and ^2^E states to a higher-lying ^4^T_2_ electronic term, from where the system can relax to the ^4^A_2_ ground state [16]. Increasing the ligand-field splitting, however, raises the energy of the ^4^T_2_ term, which in turn partially quenches nonradiative relaxation and increases the phosphorescence efficiency in [Cr(ddpd)_2_]^3+^, see Figure 1b.

The phosphorescence efficiency of [Cr(ddpd)_2_]^3+^ was found to increase even further in deuterated solvents and by deuteration of the complex itself, reaching a record quantum yield of 30% [17]. Furthermore, the phosphorescence emission wavelength is sensitive to a redshift upon applying pressure on the complex both in its solid state and in solution [18], while the emission could be blueshifted into the visible (red) spectrum by replacing the NMe bridges in the ddpd ligand with CH_2_ units in the [Cr(bpmp)_2_]^3+^ complex (bpmp = 2,6-bis(2-pyridylmethyl)pyridine) [19]. Similar large quantum yields and emission wavelengths have now also been achieved for chromium(III) complexes with dqp ligands (dqp = 2,6-di(quinolin-8-yl)-pyridine) [20,21], in which the outer NMe−pyridine units of the ddpd ligand are replaced by quinoline units. Furthermore, the emission wavelengths could be shifted far into the near-infrared II region to 1067 nm by using anionic dpc ligands (dpc = 3,6-di-tert-butyl-1,8-di(pyridine-2-yl)carbazolato), although [Cr(dpc)_2_]^+^ exhibited phosphorescence only at 77 K, while emission was quenched at room temperature [22]. Using the isoelectronic vanadium(II) in place of chromium(III) does not produce a strong enough ligand field, as [V(ddpd)_2_]^2+^ is not luminescent [23]. Vanadium(III) complexes of ddpd, in contrast, are emissive [24,25]; the different $d^{2}$ electron configuration of the vanadium(III) center, however, gives rise to a completely different phosphorescence mechanism [26].

Despite the spectacular emissive properties of [Cr(ddpd)_2_]^3+^, little detail is known about the actual mechanism that leads to the photoluminescence in [Cr(ddpd)_2_]^3+^. Transient absorption spectroscopy (TAS) after 440 nm excitation into the ^4^T_2_ state exhibited a monoexponential decay of the experimental signal with a time constant of 3.5 ps, which was first ascribed to the relaxation of the emissive doublet states without the ability to discern any intermediary steps [27]. A later interpretation of the experiments suggested that intersystem crossing from the quartet to the doublet manifold occurred on a time scale faster than 200 fs, and the previous observed picosecond time constant was rather due to vibrational cooling and internal conversion between the doublet states [28]. Step-scan Fourier transform infrared (FTIR) spectroscopy observed signals redshifted compared to the ground state in the long-lived electronically excited states [29]. The redshift is indicative of geometrical distortions compared to the ground state and was ascribed to a distortion along one of the bonding axes of the ligands due to the population of a ^2^T_1_ microstate rather than a ^2^E component. The difficulties to obtain a more detailed picture of the photodynamics arise likely from the excited-state mechanism that includes only metal-centered states. Due to the similar electronic properties of metal-centered states, such processes can be difficult to discern, compared to the more frequent studies of processes featuring both metal-centered states and electronic states of charge transfer character.

Computational chemistry can be used to gain more insight into photoluminescence mechanisms, by investigating the processes occurring in the excited states of emissive transition-metal complexes after photoexcitation [30]. While experimental studies of [Cr(ddpd)_2_]^3+^ and other emissive chromium(III) complexes have been complemented by static quantum chemical calculations of the electronic structures at their ground-state geometries [15,18,19,20,21], a dynamical picture of the excited-state reactions is so far missing. Thus, in this study, we set out to simulate the phosphorescence mechanism of the molecular ruby [Cr(ddpd)_2_]^3+^ in aqueous solution using non-adiabatic dynamics. We find that after photoexcitation to the ^4^T_1_ states, the main reaction pathways drives the system via the ^4^T_2_ to the long-lived ^2^T_1_ and ^2^E states within 2 ps. From the ^4^T_1_ term, a significant amount of electronic state population initially also transfers to the ^2^T_2_ states; however, it quickly returns to the ^4^T_2_ state to follow the main pathway. These electronic processes are driven by an increase in the Cr–N bond lengths and are accompanied by a twisting of the central, trans-coordinated pyridine units of the ddpd ligands. From the ^2^T_1_ and ^2^E states populated during the simulation, the system can easily relax to the doublet minima from where emission occurs.

## 2. Materials and Methods

### 2.1. Electronic Structure Method

The quartet ground state of [Cr(ddpd)_2_]^3+^ in water was optimized with unrestricted Kohn–Sham density functional theory using the B3LYP functional [31] and ZORA-def2-SVP basis sets [32]. Calculations were performed using the ORCA4.2.1 program package [33]. Solvent effects of water were modeled using the conductor-like polarizable continuum model (C-PCM). The D3BJ correction was used to include dispersion effects [34]. Scalar relativistic effects were included using the ZORA Hamiltonian [35]. For the self-consistent-field (SCF) calculations, the resolution-of-identity approximation (RIJCOSX) [36] with def2/J auxiliary basis sets, the tight SCF convergence criteria (TightSCF), and the Grid4 integration grid were used. A frequency calculation was conducted to compute the normal modes at this geometry as well as confirm its identity as a minimum through the absence of imaginary frequencies.

The $d^{3}$ open-shell electron configuration of [Cr(ddpd)_2_]^3+^ gives rise to many close-lying singly and doubly excited configurations that can contribute to the wave functions of excited electronic states. Thus, to describe the ground and excited electronic states of [Cr(ddpd)_2_]^3+^ adequately, we employed multiconfigurational methods [37,38] using the OpenMolcas program package version v.18.0.o180122-0800 [39]. For this, state-averaged complete-active-space self-consistent field (CASSCF) and multistate complete-active-space second-order perturbation theory (CASPT2) [40] calculations were performed using the ANO-RCC-VDZP basis set [41,42] at the optimized quartet ground-state geometry. Different active spaces including the chromium-*d* orbitals as well as ligand-centered π and $\pi^{*}$ orbitals were tested for different numbers of quartet and doublet states. The solvent effects of water were modeled using the polarizable continuum model (PCM). To speed up the integral calculations, the Cholesky decomposition (RICD) was employed [43]. To avoid intruder states, CASPT2 calculations were conducted using an imaginary level shift of 0.1 a.u. [44]. The effect of the empirical IPEA shift [45,46] on CASPT2 vertical excitation energies was investigated using shift values of 0 and 0.25 a.u. (see Appendix A); here, only results obtained with a shift value of 0.25 a.u. are discussed. Spin–orbit couplings between all states were calculated using the atomic mean-field (AMFI) approximation [47] in the restricted-active-space state interaction (RASSI) program in OpenMolcas.

### 2.2. Nonadiabatic Dynamics Simulations

The nonadiabatic dynamics were started at quartet ground state geometry of [Cr(ddpd)_2_]^3+^ and executed using the trajectory surface hopping (SH) method SHARC [48,49,50] on potential energy surfaces parametrized with a linear vibronic coupling (LVC) model [51,52,53]. The LVC parameters were determined for the lowest-lying eight doublet and seven quartet states using a combination of CASSCF and CASPT2 calculations. To this aim, the CASSCF and CASPT2 levels of theory were employed using a (5,3) active space and an IPEA shift of 0.25 a.u. (see Section 2.1). The interstate and intrastate coupling elements were obtained numerically from the CASSCF calculations on geometries displaced by ±0.05 units from the quartet ground-state geometry for each of the 231 vibrational normal modes of the molecule. Intrastate coupling constants were obtained as numerical gradients; interstate coupling constants were derived from the change in the wave function overlaps [54]. Spin–orbit couplings were calculated at the quartet ground-state geometry at the CASSCF level of theory. Finally, vertical excitation energies were calculated at the quartet ground-state geometry at the CASPT2 level of theory.

The SH/LVC dynamics calculations were started from 1000 trajectories generated with a Wigner ensemble of geometries and velocities at the quartet ground-state geometry [55]. All trajectories were initialized for states within the second absorption band (3.0–4.2 eV) based on their oscillator strength [56]. Initial states were chosen in the adiabatic representation, leading to trajectories started in the $Q_{3}$ (1), $Q_{4}$ (21), $Q_{5}$ (656), $Q_{6}$ (179), and $Q_{7}$ (143) adiabatic quartet states—the ground state being denoted as $Q_{1}$. The trajectories were then propagated using the SHARC approach and employing the pySHARC driver [50] for 10 ps, using a nuclear time step 0.5 fs and an electronic time step of 0.02 fs within the local diabatization method [57]. An energy-based decoherence correction with a constant of C=0.1 a.u. was used [58]. During the surface hops, the kinetic energy was adjusted by rescaling the velocity vectors. Surface hopping probabilities were approximated using the wave function overlaps [59].

## 3. Results and Discussion

### 3.1. Electronic Structure

[Cr(ddpd)_2_]^3+^ features a chromium(III) metal center with a $d^{3}$ electron configuration, which gives rise to a quartet ground state, which we labeled as ^4^A_2_ due to the pseudo-octahedral geometry of the complex. The pseudo-octahedral geometry leads to sets of nearly degenerate orbitals, i.e., the three nonbonding $t_{2g}$ and two antibonding $e_{g}^{*}$ orbitals at the chromium center, which further give rise to nearly degenerate electronic states [18]. Such complicated electronic structures are best described using multiconfigurational approaches [37,38]. Here, we thus employed a CASSCF/CASPT2 ansatz with an active space comprising the five 3d orbitals as well as two π and $\pi^{*}$ orbitals in order to describe both metal-centered states as well as states involving excitations at the ligands. The resulting energies and state composition of the lowest-lying quartet and doublet excited states are shown in Figure 2.

As can be seen, the ^4^A_2_ ground state (violet line/box) of [Cr(ddpd)_2_]^3+^ in water is followed by three almost degenerate quartet states between 2.53 and 2.71 eV that correspond to the ^4^T_2_ term (orange). The states are all described by linear combinations of two or three configurations that feature excitations from a $t_{2g}$ to an $e_{g}^{*}$ orbital, showcasing the typical multiconfigurational character of metal-centered states in open-shell complexes. The states of the ^4^T_2_ term are followed by another triad of almost degenerate states between 3.62 and 3.79 eV that correspond to the ^4^T_1_ term (red). These states are also dominated by a $t_{2g}\to e_{g}^{*}$ excitation; however, their wave functions also possesses already significant ${\left(t_{2g}\right)}^{2}\to {\left(e_{g}^{*}\right)}^{2}$ double excitation contributions of 13–15%. Following the ^4^T_2_ term, CASPT2 predicts two ligand-centered states (brown) at 4.05 and 4.56 eV described by the $\pi \to \pi^{*}$ excitation in the ddpd ligands. Above these states, further metal-centered (yellow) and ligand-centered (brown) states follow, for which we only show their energies in the middle of Figure 2; further information in their state character is given in Appendix A.

Previous computational studies of [Cr(ddpd)_2_]^3+^ used the single-reference time-dependent density functional theory (TDDFT) approach to describe quartet excited states contributing to the absorption spectrum [15]. Using the B3LYP functional, the lowest-energy absorption band of [Cr(ddpd)_2_]^3+^ in water and acetonitrile at ∼435 nm (2.85 eV) was ascribed to an excitation to the ^4^T_2_ term as well as ligand-to-metal charge transfer (LMCT) excitations [15]. Our CASPT2 calculations find the ^4^T_2_ state close to the experimental absorption band, blue-shifted by 0.1–0.3 eV, see Figure 3a,b. A study using the multireference method NEVPT2 with an active space restricted to target only metal-centered states [18] also found states of the ^4^T_2_ term at 2.8–3.0 eV (442–413 nm), supporting the assignment of the 435 nm band to the ^4^T_2_ states. However, our CASPT2 calculations find no LMCT states at such low energies, thus questioning the additional assignment of the 435 nm band to LMCT states as well. The first non-metal-centered states in CASPT2 are predicted at 4.05 eV (306 nm) and involve ligand-centered excitations. A similar mismatch between TDDFT and CASPT2 predictions for low-lying excited states of open-shell transition-metal complexes was discussed for the VCl_3_(ddpd) complex. In both cases, it is likely that the disagreement is related to the incapability of standard density functionals in linear-response TDDFT to correctly describe charge-transfer states as well as higher-order excitations [25,26].

In the doublet manifold, CASPT2 predicts five electronic states between 1.85 and 2.01 eV (670–617 nm; blue line/box in Figure 2), belonging to the ^2^E and ^2^T_1_ electronic terms. Following Ref. [18], we ascribe the two states where all three $t_{2g}$ orbitals are singly occupied to the ^2^E term (light blue), while we classify the other three states that feature configurations with doubly occupied $t_{2g}$ orbitals as belonging to the ^2^T_1_ term. While in ideal octahedral symmetry, the lowest doublet electronic state is given by the ^2^E term, the energetic splitting of the states resulting from the pseudo-octahedral symmetry of [Cr(ddpd)_2_]^3+^ makes a component of the ^2^T_1_ term the lowest doublet state. The ^2^E and ^2^T_1_ terms are followed by three states between 2.91 and 3.02 eV (426–410 nm), which correspond to the ^2^T_2_ term (green) and also feature ${\left(t_{2g}\right)}^{3}$ electron configurations. Above these states lies a single ligand-centered doublet state (brown) and multiple metal-centered doublet states (yellow) derived from $t_{2g}\to e_{g}^{*}$ excitations (see Appendix A for the state characterization). Single-crystal absorption spectra of [Cr(ddpd)](BF_4_)_3_ evidenced three low-intensity signals at 697 (1.77 eV), 736 (1.68 eV), and 776 nm (1.59 eV) that were ascribed to spin-forbidden transitions from the ^4^A_2_ ground state to the terms ^2^T_2_, ^2^T_1_, and ^2^E terms, respectively [15]. Given the large energy gap between the five ^2^E and ^2^T_1_ states and the three ^2^T_2_ states as observed from our CASPT2 calculations, it seems more likely, however, that the three bands correspond only to states from the ^2^E and ^2^T_1_ terms, as the ^2^T_2_ lies at energies well above the observed single-crystal absorption bands, see Figure 3a,c.

### 3.2. Setup of the LVC Model

Based on the above quantum chemical results, we set out to parametrize the LVC model potentials that allow us to investigate the photodynamics of [Cr(ddpd)_2_]^3+^ using trajectory surface hopping. We decided to excite to the ^4^T_1_ states in the second absorption band and based on their energy, we include only the states of the ^4^A_2_, ^4^T_2_, ^4^T_1_, ^2^E, ^2^T_1_, and ^2^T_2_ terms, i.e., seven quartet states and eight doublet states. Noting that these states were all of MC nature, it is possible to contract the size of the active space to a (5,3) active space that includes only the three electrons in the five 3d orbitals. Such smaller active space introduces only minor changes (<0.1 eV) in the energies of these states (Appendix A), justifying the choice. However, we noted that the LVC parametrization using CASPT2 leads to problems in the dynamics, such as negative energies and nonphysical large geometrical displacements. In previous applications [60,61], such problems were solved by removing low-frequency normal modes or specific electronic states from the LVC model. No such problems were observed in the simulation of VCl_3_(ddpd) [26] where the LVC model was parametrized from CASSCF calculations.

For [Cr(ddpd)_2_]^3+^, CASSCF is not sufficient to parametrize the LVC model because the differences between CASSCF and CASPT2 excitation energies are sizable (0.2–0.6 eV), regardless of using the larger (9,5) or the smaller (5,3) active space. As a solution, here, we use a combination of CASSCF and CASPT2 properties in order to set up the LVC model. In particular, we use CASSCF to compute the interstate and intrastate coupling parameters as well as the spin–orbit couplings, but we use CASPT2 to obtain accurate vertical CASPT2 excitation energies at the ground-state geometry. In this way, the shape of the potential energy surfaces in the LVC model is described at the CASSCF level, while the energetic separation (at the reference geometry) is controlled using CASPT2. This approach is reminiscent of static explorations of potential energy surfaces where CASSCF and CASPT2 are combined by optimizing geometries at the CASSCF level and recalculating energies at the CASPT2 level [62]. Notably, the resulting dynamics are free of unphysical negative energies or unrealistic large nuclear displacements, at least for this system. We note that photodynamics studies of [Fe(NCH)_6_]^2+^ have also combined different levels of theory to setup LVC models for surface-hopping simulations, [63,64] although in a different way. There, singlet and triplet electronic states were described using TDDFT based on a restricted Kohn–Sham ground-state wave function, quintet states were described using unrestricted density functional theory, and the spin–orbit couplings between all states were approximated from corresponding CASSCF/CASPT2 calculations.

### 3.3. Excited-State Dynamics

Exciting [Cr(ddpd)_2_]^3+^ from the ^4^A_2_ ground-state geometry to the second absorption band leads to the excited-state dynamics shown in Figure 4. The initial excitation and propagation of the electronic states in the surface-hopping simulations is performed in the adiabatic representation [65]. In this representation, states are characterized by their energetic ordering and spin multiplicity. Adiabatic states can change their electronic character along the dynamics, as the electronic state composition can change at different geometries or when the energetic ordering of the states changes. For the analysis, however, we use a diabatic representation [66], where the electronic states are classified according to their character that does not change. As reference states for the diabatic representation, we use the (adiabatic) states from the ground-state geometry, where we continue to use the octahedral term notation introduced in Section 3.1.

It is important to realize that our analysis simplifies the description of the dynamics occurring in the simulation. When a trajectory moves away from the starting geometry, its electronic state composition changes and adopts a new character different from any of the electronic states at the starting geometry. The new electronic configuration can be expressed as a linear combination of a set of (diabatic) reference states. Trajectories do not remain in any pure reference states, and the change of the electronic wave function in the dynamics usually occurs gradually. This also holds true in light of the sudden character of the hop in the surface-hopping simulations. Surface hops only change the active state, i.e., the state whose gradient the trajectory follows, but not the total electronic wave function of the trajectory.

The time evolution of the diabatic electronic state populations of the surface-hopping simulations is shown for the first 1 ps of simulation time and the full 10 ps of simulation time as thin lines in Figure 4a. As can be seen, the dynamics start in the ^4^T_1_ states (red line). Immediately after, a sizable amount of population is transferred from the ^4^T_1_ to the ^2^T_2_ states (green line) that reaches ca. 22% after 20 fs. This ultrafast population transfer is likely due to the prompt intersystem crossing [67], a purely electronic effect due to the large spin–orbit coupling of the chromium metal center, and does not involve or require any nuclear motion to proceed. The ^4^T_1_ and ^2^T_2_ curves then show mirroring oscillations that fade out around a simulation time of 1 ps, suggesting an equilibrium of back-and-forth intersystem crossing between the states of these two terms.

A slightly slower rise than that of the ^2^T_2_ states is displayed by the ^4^T_2_ states (orange line) that reaches ca. 25% after 220 fs, peaks at 630 fs with 27%, and decreases over the long simulation time to 11% at 10 ps. Subsequently, the population also accumulates in the low-energy doublet terms ^2^E and ^2^T_1_. We show here the combined populations ^2^E + ^2^T_1_ (blue line) to simplify establishing a reaction mechanism. The individual populations of both terms behave in a synchronous manner as shown in Appendix A. The populations only differ in their amplitude that follows the 2:3 ratio of the components of the ^2^E and ^2^T_1_ terms. The ^2^E + ^2^T_1_ population grows fast to ca. 80% after a 2 ps simulation time, after which only a small growth remains to drive the population to ca. 85% after 10 ps. Finally, a small part of the population also reaches the ^4^A_2_ ground state, which reaches ca. 1% after 10 ps, similar to the populations of the ^2^T_2_ and ^4^T_1_ terms after 10 ps.

### 3.4. Mechanism

The analysis of the electronic state populations already suggests a number of possible reactions that could contribute to the deactivation mechanism of the excited-state dynamics of [Cr(ddpd)_2_]^3+^. For example, the mirrored oscillatory behavior of the ^4^T_1_ and ^2^T_2_ populations indicates back-and-forth reactions between these terms. However, building a complete mechanism based solely on analyzing the time evolution of the electronic state populations is not straightforward, in particular when a large number of electronic states and a consequently larger number of possible reactions are involved. In such cases [26], it is helpful to analyze the electronic character of the individual trajectories in order to identify the frequent—and thus important—electronic processes that make up the mechanism. Such an analysis (Figure 4b) shows the diabatic character of the total electronic wave function for a set of 50 random trajectories for the first 4 ps of simulation time. Each trajectory is represented by a colored line that denotes the diabatic character. The trajectories are only colored if the weight of the dominant diabatic state contribution exceeded 80% of the total electronic wave function. Blank spots denote parts of the trajectory where there was no contribution larger than 80%. This was a larger threshold than we used in previous works on VCl_3_(ddpd) [26] (60–70%) and [Fe(cpmp)_2_]^2+^ [68] (40–50%; cpmp = 6,2″-carboxypyridyl-2,2′-methylamine-pyridyl-pyridine), showing that in the excited-state dynamics of [Cr(ddpd)_2_]^3+^ the trajectories stay in electronic states more similar to the states at the Franck–Condon geometry than in the other two examples noted. Note, however, that we collect the almost-degenerate electronic states corresponding to the components of the octahedral electronic term in this analysis, i.e., the contribution of the individual diabatic state components is usually lower.

As can be seen in Figure 4b, trajectories starting in the ^4^T_1_ states (red) either go to the ^2^T_2_ states (green)—from where they return to the ^4^T_1_—or go to the ^4^T_2_ states (orange). From the ^4^T_2_ states, almost all trajectories transfer to the ^2^E + ^2^T_1_ states (blue), while a few trajectories also relax to the ^4^A_2_ ground state (violet). In the ^2^E + ^2^T_1_ states, trajectories oscillate for short times back and forth to the ^4^T_2_ states. From these observations, we constructed a reaction network to describe the mechanism of the simulated excited-state dynamics. From the reaction network, we fitted populations curves to the simulated population curves from the time evolution of the electronic state populations, whereby a good agreement between fitted and simulated curves suggests that a suitable reaction network is found.

Using only the reactions described in the previous paragraph does not yield fits that describe the simulated populations curves adequately, as is shown in Appendix A. Thus, we added modifications to the reaction network. In particular, as small parts of the population remain in the ^2^T_2_ and ^4^T_1_ states, we added ^2^T_2_^cold^ and ^4^T_1_^cold^ states in the fits, where population could accumulate. In addition, the ^4^T_2_ → ^2^E + ^2^T_1_ intersystem crossing is better modeled as a combination of a fast and a slow process [61]. Finally, adding the reverse intersystem crossing from the ^2^E + ^2^T_1_ to the ^4^T_2_ states deteriorates the fits, so this process was removed from our model. All modifications are discussed in more detail in Appendix A.

The final reaction network and its time constants are displayed in Figure 4c. Fits based on these reactions are shown as thick curves in Figure 4a, which closely follow the electronic-state populations from the simulations (thin lines), thus confirming that the reaction network can adequately describe the simulated reactions. As can be seen in Figure 4c, the initial depopulation of the ^4^T_1_ states to the ^2^T_2_ and ^4^T_2_ states occurs on a timescale of ∼350 and ∼450 fs, respectively. Reverse intersystem crossing from the ^2^T_2_ to the ^4^T_1_ states is even faster on a ∼100 fs time scale, while the population of the ^2^T_2_^cold^ and ^4^T_1_^cold^ states occurs on a slow 10–20 ps time scale. From the ^4^T_2_ term, the ^2^E + ^2^T_1_ states are reached on a fast <150 fs reaction and a slow one (∼30 ps). The fast, <150 fs time-scale intersystem crossing is in line with the more recent interpretation [28] of the TAS experiments of Ref. [27]. Additionally, a small amount of the population relaxes from the ^4^T_2_ states to the ^4^A_2_ ground state on a >10 ps time scale.

The excited-state dynamics mechanism can be further simplified by calculating the relative contributions of the (most important) reaction pathways from the fitted time constants as shown in Figure 4d. According to this analysis, ca. 55% of the initial population from the ^4^T_1_ states undergo intersystem crossing first to the ^2^T_2_ states and then back to the ^4^T_1_ states. From there, almost all of the population (∼98%) then moves via the ^4^T_2_ states to the ^2^E + ^2^T_1_ states—only 1% end up in the ^4^A_2_ ground state. Overall, excitation of [Cr(ddpd)_2_]^3+^ to the ^4^T_1_ states brings the system very efficiently to the ^2^E + ^2^T_1_ states as the simplified mechanism in Figure 4d highlights. However, the individual pathways on which the system traveled are usually more complicated as revealed by the individual trajectories shown in comparison in Figure 4b.

### 3.5. Nuclear Motion

In order to identify the most important nuclear coordinates that are excited during the dynamics, we performed a normal-mode analysis [69,70]. Figure 5 shows that the largest distortion occurred in the Cr–N bond lengths r_Cr–N_ between the chromium central metal and the ligating N atoms. Notably, there is a qualitative difference between the distances to the central pyridine units in the ddpd ligand (py_2_ and py_3_) and the peripheral pyridine units (py_4_–py_7_); see Figure 5a for the numbering of the nitrogen atoms/pyridine units.

Starting from an initial ground-state value of 2.05 Å, the r_Cr–N2_ central bond length quickly increases to reach ca. 2.25 Å after 110 fs, before it decreases back to the starting value at 280 fs (Figure 5b). These oscillations repeat with a similar period of ca. 260–280 fs, with concomitant increasing and decreasing of the r_Cr–N2_ central bond length by ca. 0.2 Å. The r_Cr–N4_ peripheral bond length has a similar starting value of 2.03 Å, which also increases rapidly to ca. 2.25 Å after 80 fs (Figure 5c). It subsequently decreases to a value of ca. 2.15 Å around which it fluctuates, however, with rather damped, nonperiodic oscillations.

The increase of both Cr–N bond lengths can be rationalized by the electronic states that are populated in the dynamics. The system is excited to the ^4^T_1_ state which brings the system from the ${\left(t_{2g}\right)}^{3}$ ground-state electron configuration to a ${\left(t_{2g}\right)}^{2}{\left(e_{g}^{*}\right)}^{1}$ configuration (Figure 2), where one electron is transferred from a nonbonding $t_{2g}$ to an antibonding $e_{g}^{*}$ orbital. Such states are stabilized by increasing the metal-ligand bond lengths, which decreases the antibonding character of the $e_{g}^{*}$ orbitals. Thus, population of the ^4^T_1_ states induces a lengthening of the Cr–N bonds. From the ^4^T_1_ states, the system traveles partially to another state of the ${\left(t_{2g}\right)}^{2}{\left(e_{g}^{*}\right)}^{1}$ configuration, i.e., ^4^T_2_, leading to a similar nuclear motion. The other part of the population transfers to the ^2^T_2_—with a ${\left(t_{2g}\right)}^{3}$ configuration, similar to the ground state—and back again to the ^4^T_1_ state. It was not clear whether this back-and-forth reaction is reflected in any Cr–N bond length evolution. For example, in the r_Cr–N2_ central bond length evolution, there is no different behavior in the first 1 ps, while there is still a significant population in the ^2^T_2_ states, and at later times, while the ^2^T_2_ population is almost negligible. Thus, trajectories that switch from the initially excited ^4^T_1_ states to the ^2^T_2_ states could just follow the initial momenta towards larger bond lengths induced by the gradients of the ^4^T_1_ states.

The behavior of the r_Cr–N4_ peripheral bond lengths in [Cr(ddpd)_2_]^3+^ is similar to that of the V–N bond lengths in the photodynamics of VCl_3_(ddpd) [26]. Its dynamics was also dictated from the population and depopulation of antibonding $e_{g}^{*}$ orbitals, even if VCl_3_(ddpd) features a $d^{2}$ electron configuration with different electronic states than [Cr(ddpd)_2_]^3+^. Interestingly, however, the photodynamics of VCl_3_(ddpd) did not show any coherent oscillatory motions such as those of the r_Cr–N2_ central bond lengths in [Cr(ddpd)_2_]^3+^. Moreover, for VCl_3_(ddpd), a lowering of the angles between trans-coordinated pyridine rings were observed. These changes were also rationalized by the population of electronic states with occupied $e_{g}^{*}$ orbitals, as lowered trans angles deform the complex further away from an idealized octahedral geometry, where the antibonding character of the $e_{g}^{*}$ orbitals is the strongest. Such changes in the trans angles are not found in the dynamics of [Cr(ddpd)_2_]^3+^, as we show exemplarily in Figure 5d) for the γ_N_2_–Cr–N_3__ angle between the central ligating pyridine units and the chromium atom, which, throughout the simulation time, stays around 175–177${}^{\circ }$.

Finally, we noticed an interesting motion in the dynamics that affects the relative orientation of the trans-coordinated pyridine rings in the ddpd ligands. While the rings of the central pyridine units py_2_ and py_3_ are aligned in a rather coplanar orientation at the ground-state geometry, the dynamics induces twisting motions that brought the ligands towards a more diagonal orientation. To quantify this, we calculated the angle Φ_py_2_–py_3__ between the averaged planes through the atoms of the pyridine rings py_2_ and py_3_ as shown in Figure 5e). As can be seen, the initial angle of 161${}^{\circ }$ decreases during the dynamics and oscillates between 130 and 150${}^{\circ }$. In contrast to the central pyridine units, the rings of trans-coordinated peripheral pyridine units, e.g., py_4_ and py_5_, are aligned almost perpendicular at the ground-state geometry. They are also brought to a more diagonal orientation in the dynamics, as characterized by an angle Φ_py_4_–py_5__ that decreases from 81${}^{\circ }$ at the initial ground-state geometry towards values of 50–80${}^{\circ }$ during the simulation, see Figure 5f. This motion decreasing the angles of all trans-coordinated ligands is also shown in Appendix A. For comparison, the angles between the cis-coordinated pyridine units do not change as much during the dynamics as exemplified by the Φ_py_3_–py_6__ angle in Figure 5g. This angle changes from an initial value of 41${}^{\circ }$ to values of 30–50${}^{\circ }$ during the dynamics. Based on these observations, it could be interesting to test the effect of substitution of the ddpd ligand which may hinder or facilitate the twisting of the pyridine units, e.g., by replacing the methyl group at the amine bridges with smaller or larger substituents, such as hydrogen atoms or tert-butyl groups, respectively.

### 3.6. Near-Infrared Emission

The extraordinary feature of [Cr(ddpd)_2_]^3+^ is its near-infrared emission occurring with a quantum yield of 11% in water at room temperature [15]. Our dynamics simulation reveals that the excitation of the ^4^T_1_ states ultimately leads to the population of the low-lying ^2^E + ^2^T_1_ states, which are thus assigned as responsible for the emission. According to Kasha’s rule [71], emission usually occurs from the lowest minimum-energy geometry. As the emission spectrum of [Cr(ddpd)_2_]^3+^ shows two bands centered at 740 and 778 nm, see Figure 3a, emission occurs from two distinct electronic states. These states can either be located at different energetic minima, occur from two different electronic states, or both.

In order to investigate the emission feature of [Cr(ddpd)_2_]^3+^, we set out to locate the lowest-lying doublet minima using our LVC model potentials. For this, we performed excited-state geometry optimizations starting from all 846 trajectories that were in a doublet state after 10 ps. All optimizations laed to very similar geometries of the lowest-lying doublet state D_1,min_. The optimized geometries are shown in Appendix A, and we show a superposition with the ground-state geometry in Figure 3d). The doublet minima geometries and the ground-state geometry mainly differ in the Cr–N bond lengths and the twisting of the trans-coordinated pyridine units. The Cr–N bond distances are ca. 2.10 Å at the D_1,min_ geometry, while the trans-angles amount to Φ_py_2_–py_3__$=141^{\circ }$ and Φ_py_4_–py_5__$=61^{\circ }$, respectively. These distances and angles are consistently reached (or traversed) by the trajectories throughout the dynamics simulation—see Figure 5—suggesting that the D_min_ minima are easily reached.

At all optimized geometries, the lowest-lying doublet state D_1,min_ is found at an excitation energy of 1.66 eV (746 nm), i.e., blue-shifted by 0.05 eV compared to the experimentally observed lower-energy emission band at 1.60 eV (775 nm), see Figure 3a,c. At these geometries, the second-lowest doublet state D_2,min_ is found at an energy of 1.69 eV (733 nm), with a similar blue-shift of 0.02 eV compared to the experimentally observed higher-energy emission band at 1.67 eV (740 nm). Our results thus suggest that the two experimentally observed emission bands result from the two lowest-lying components of the ^2^E + ^2^T_1_ states, which, based on the ordering of the doublet states at the ground-state geometry, correspond to the lowest-energy components of both terms.

## 4. Conclusions

The complex [Cr(ddpd)_2_]^3+^—known as the molecular ruby—displays exceptionally high luminescence quantum yields for a 3d transition-metal complex due to the combination of electronic and geometric properties of the ddpd ligand [15]. To gain insight into the photophysical processes responsible for these properties, here, we simulated the nonadiabatic dynamics after photoexcitation of [Cr(ddpd)_2_]^3+^ in water using trajectory surface hopping on linear vibronic coupling (LVC) potentials along all 231 nuclear degrees of freedom. The complex electronic structure resulting from the open-shell chromium(III) metal center requires a multiconfigurational description of the potentials, which in this case is achieved with a combination of CASSCF and CASPT2 approaches.

The photodynamics mechanism of [Cr(ddpd)_2_]^3+^ is summarized in Figure 6. After exciting the system into the ^4^T_1_ states in the second absorption band, we observed a bifurcation of the dynamics. Half of the electronic population initially undergoes rapid intersystem crossing to the ^2^T_2_ states; however, it returns back to the ^4^T_1_ states within 1 ps. All population from the ^4^T_1_ states then relaxes to the ^4^T_2_ states on a <500 fs time scale and undergoes further intersystem crossing to the nearly degenerate ^2^E and ^2^T_1_ states. The intersystem crossing takes place on a fast <150 fs time scale—moving ca. 80% of the population within 2 ps into the low-lying doublet states—and a slower ∼30 ps process. The nuclear motion involved in the electronic processes mainly involves an increase of the Cr–N bond lengths. This is due to the population of electronic states in which an electron is excited from the nonbonding $t_{2g}$ orbitals to the antibonding $e_{g}^{*}$ orbitals. The increase of the Cr–N bond lengths is accompanied by a twisting of the trans-coordinated pyridine units in the ddpd ligands. Both motions lead the system close to the region of the doublet minimum, suggesting that the emissive doublet states can be easily populated.

## Figures and Tables

**Figure 1 molecules-28-01668-f001:**
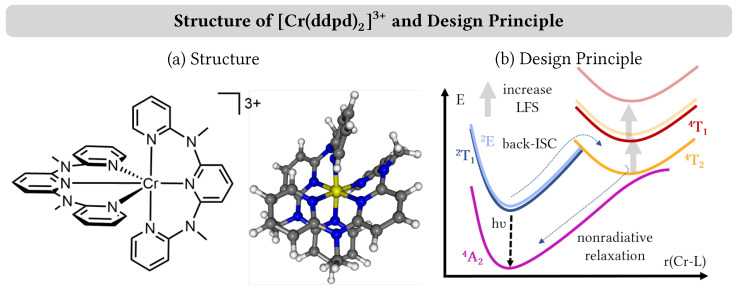
(**a**) Structure of [Cr(ddpd)_2_]^3+^. (**b**) Design principle of molecular rubies: by increasing the ligand-field splitting (LFS) in octahedral chromium(III) complexes, back-intersystem crossing (ISC) from the low-lying ^2^E + ^2^T_1_ to the ^4^T_2_ states at larger chromium–ligand bond lengths and subsequent nonradiative relaxation to the ^4^A_2_ ground state is quenched, resulting in increasing luminescence yield. The gray arrows denote the effect of increased LFS.

**Figure 2 molecules-28-01668-f002:**
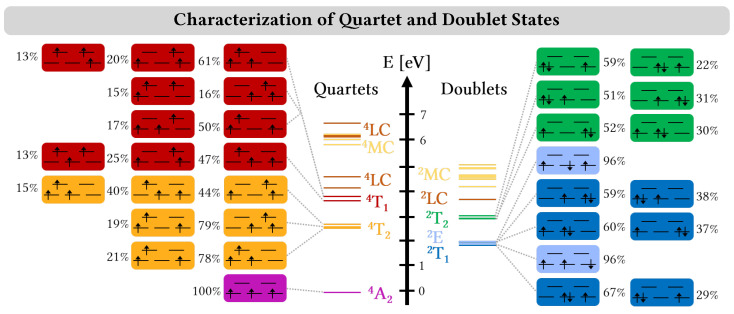
Energies and wave function characterization of quartet and doublet states of [Cr(ddpd)_2_]^3+^ in water calculated at the CASPT2 level of theory. The lowest-lying states are metal-centered (MC) states and have been labeled using octahedral electronic term notation. States derived from the same terms are shown in the same color. Higher-lying MC and ligand-centered (LC) states are labeled as such in the middle of this figure and not distinguished further. Wave function characterization depicts the all configuration with weights >10% in the MS-CASPT2 wave function, where the orbitals depict the three nonbonding $t_{2g}$ (lower levels) and two antibonding $e_{g}^{*}$ (upper levels) orbitals. More information about the further MC and LC states is given in Appendix A.

**Figure 3 molecules-28-01668-f003:**
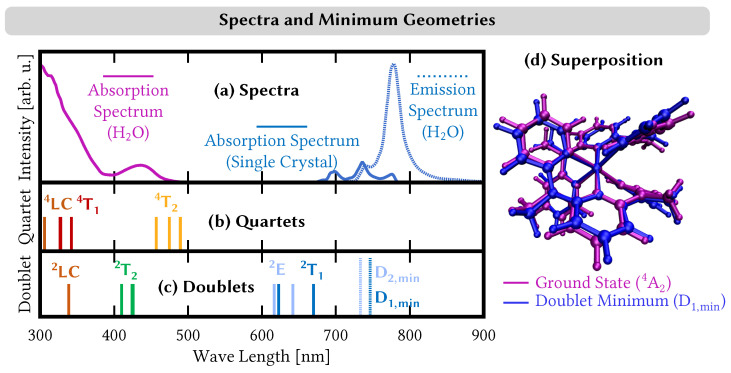
(**a**) Experimental absorption spectra [15] of [Cr(ddpd)_2_]^3+^ in water and in single-crystal form as well as emission spectrum of [Cr(ddpd)_2_]^3+^ in water. Intensities scaled arbitrarily. (**b**/**c**) Stick spectrum of quartet/doublet states computed using CASPT2 in water. Dashed lines corresponds to the energy of LVC-optimized doublet minimum (D_1,min_) and the second-excited doublet state at this geometry (D_2,min_). (**d**) Superposition of geometries of the ground state (^4^A_2_) and the lowest doublet-state minimum (D_1,min_).

**Figure 4 molecules-28-01668-f004:**
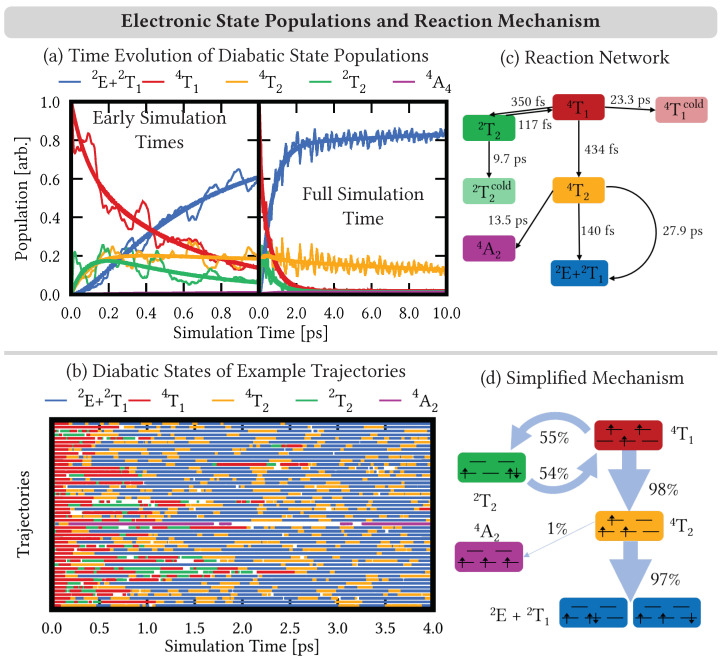
(**a**) Time evolution of the diabatic electronic state populations (thin lines) of states combined into terms according to Figure 2. Thick lines represent fitted population curves according to the reaction network in (**c**). (**b**) Time evolution of 50 example trajectories. The trajectories are colored according to their dominant diabatic state contribution if it exceeds 80% of the wave function character. (**d**) Simplified reaction mechanism following the reaction network in (**c**). The thickness of the arrows connecting the electronic terms corresponds to the contribution of their pathway to the mechanism.

**Figure 5 molecules-28-01668-f005:**
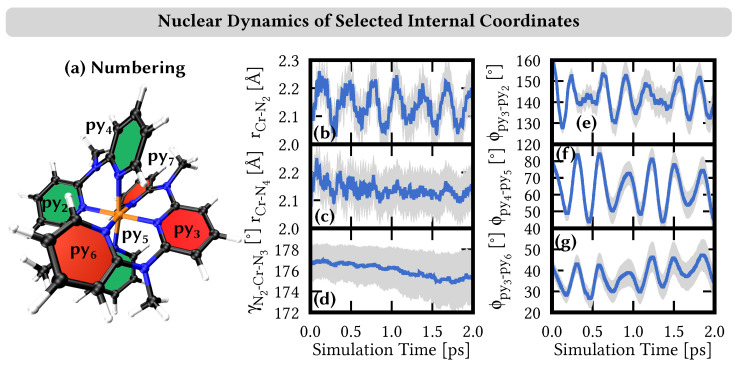
(**a**) Numbering of pyridine rings, which also corresponds to the numbering of N atoms. (**b**–**g**) Time evolution of selected internal nuclear coordinates along the simulation. Blue line denotes the average of the coordinates of 1000 trajectories, while gray areas denote the areas of standard deviation. (**b**/**c**) Cr–N bonding distances r_Cr–N_. (**d**) N${}_{2}$–Cr–N${}_{3}$ angle γ. (**e**–**g**) Angle ϕ between the averaged planes through the atoms of the pyridine rings. Additional bond lengths and angles are presented in Appendix A.

**Figure 6 molecules-28-01668-f006:**
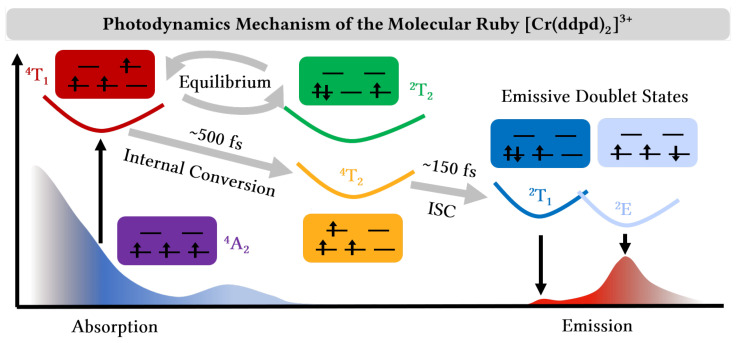
Summary of the photodynamics mechanism of the molecular ruby [Cr(ddpd)_2_]^3+^. After excitation to the ^4^T_1_ states and an initial equilibration with the ^2^T_2_ states, the system relaxes to the ^4^T_2_ states and undergoes intersystem crossing (ISC) to the ^2^T_1_ and ^2^E states from where emission occurs.

## Data Availability

Additional data and analysis are given in the Supporting Information. Raw computational data may be obtained from the corresponding authors upon reasonable request.

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
