# Peer review of "Photodynamics of the Molecular Ruby [Cr(ddpd)2]3+"

_molecules, 2023, doi:10.3390/molecules28041668_

Round 1
Reviewer 1 Report
Manuscript by Zobel et al features a theoretical analysis on the phosphorescence mechanism of the molecular ruby [Cr(ddpd)2]3+ in aqueous solution using non-adiabatic dynamics. It is shown that after excitation to the second absorption band, a relaxation cascade through metal-centered states occurs. After an initial back-and-forth intersystem crossing with higher-lying doublet states, the complex relaxes through a manifold of quartet metal-centered states to the low-lying doublet metal-centered states which are responsible for the experimentally observed emission.
The presented results will deepen the understanding of structure-function relationship of the molecular ruby [Cr(ddpd)2]3+ and help to explain luminescence characteristics in similar transition metal complexes with strong-field ligands. I therefore believe that this work can be published after some miner corrections. Below are the minor comments which will improve the quality of the manuscript:
[1] Line 97. The position of 4T1 state is not identified relative to other states earlier in the text. This state should be presented in Figure 1 (b).
[2] Line 164-168. Appropriate references for the presented statements should be provided.
[3] Line 198 and 199. The position of the 435 nm band and the first non-metal-centered states should be presented in the same units.
Reviewer 2 Report
In the manuscript of Zobel et al. computational investigations on the excited state dynamics of an emissive Cr(III) complex is reported. The calculations have been performed competently. The manuscript is very well written and structured, the scholarly presentation is of highest standard. There are only some very minor, more cosmetic comments:
# first sentence of the introduction section: "The rich photochemistry of ...". I would add also "photophysics" as in this manuscript no classical photochemical reaction is described. I am aware that there is the idea that photophysics is also photochemistry and vice versa. However, from the point of view of a chemist, chemistry is usually related to a change of the bonding situation (dissociation, isomerization, etc.). For excited coordination compounds you can observe a slight change of the bond length and angle but often no cleavage of bonds.
# second paragraph of introduction section: I can recommend the very instructive review article by Vogler in Topics of current chemistry 2001, 213, 143. It is already somewhat old. However, it nicely summarizes very systematically possible excited states of luminescent metal complexes including 3d metals in different electron configurations. This paper is always worth citing ...
# line 35: it should read copper(I)
# line 41: There is a formal issue with the formulation of [Cr^III(O^2-)6]. In this form, the formula is not correct. The square brackets [...] indicate a metal complex. Hence, three negative charges […]^3- would result. It would be more appropriate to write something like "... in analogy to emissive Cr(III) ions surrounded by six O^2- ions in solid ruby."
# line 148: delete one "were"
# line 157: "am" -> "an"
# line 266: the verb is missing in "... we a diabatic representation ..."
# line 410: I would suggest "co-planar" instead of "parallel" which is usually the term coordination chemist would use for this geometrical situation.
# last paragraph of 3.5: It would make it much easier to understand this paragraph if the authors could provide an additional schematic sketch of the structure where the geometry changes in the excited state are illustrated with arrows.
In conclusion, I recommend the acceptance of the manuscript after very minor revisions indicated above.
